# Association between Endodontic Infection, Its Treatment and Systemic Health: A Narrative Review

**DOI:** 10.3390/medicina58070931

**Published:** 2022-07-14

**Authors:** Sadia Ambreen Niazi, Abdulaziz Bakhsh

**Affiliations:** 1Department of Endodontics, Centre of Oral Clinical & Translational Sciences, Faculty of Dentistry, Oral & Craniofacial Sciences, Guy’s Dental Hospital, King’s College London, London SE1 9RT, UK; 2Department of Restorative Dentistry, Faculty of Dentistry, Umm Al-Qura University, Makkah 24381, Saudi Arabia; aabakhsh@uqu.edu.sa

**Keywords:** apical periodontitis, systemic health, cardiovascular diseases, diabetes mellitus, pregnancy, autoimmune disorder, bacteremia

## Abstract

The ‘Focal Infection Era in Dentistry’ in the late 19th and early 20th century resulted in widespread implementation of tooth extraction and limited the progress of endodontics. The theory proposed that bacteria and toxins entrapped in dentinal tubules could disseminate systemically to remote body parts, resulting in many types of degenerative systemic diseases. This theory was eventually refuted due to anecdotal evidence. However, lately there has been increased interest in investigating whether endodontic disease could have an impact on general health. There are reviews that have previously been carried out on this subject, but as new data have emerged since then, this review aims to appraise the available literature investigating the dynamic associations between apical periodontitis, endodontic treatment, and systemic health. The available evidence regarding focal infection theory, bacteraemia and inflammatory markers was appraised. The review also collated the available research arguing the associations of apical periodontitis with cardiovascular diseases, diabetes mellitus, adverse pregnancy outcome and autoimmune disorders, along with the effect of statins and immunomodulators on apical periodontitis prevalence and endodontic treatment prognosis. There is emerging evidence that bacteraemia and low-grade systemic inflammation associated with apical periodontitis may negatively impact systemic health, e.g., development of cardiovascular diseases, adverse pregnancy outcomes, and diabetic metabolic dyscontrol. However, there is limited information supporting the effect of diabetes mellitus or autoimmune disorders on the prevalence and prognosis post endodontic treatment. Furthermore, convincing evidence supports that successful root canal treatment has a beneficial impact on systemic health by reducing the inflammatory burden, thereby dismissing the misconceptions of focal infection theory. Although compelling evidence regarding the association between apical periodontitis and systemic health is present, further high-quality research is required to support and establish the benefits of endodontic treatment on systemic health.

## 1. Apical Periodontitis Aetiology

Endodontic infection is a polymicrobial infection, and the diversity of the endodontic microbiome and its host interactions presents not only a unique challenge to treatment, but also a potential risk for systemic disease in other parts of the body [1].

Chronic apical periodontitis is a dynamic sequel to root canal infection. It is driven by persistent localized inflammation within the periapical tissue that can lead to progressive bone resorption and the formation of periapical lesions. If this is left untreated, it can form a sinus tract and lead to cyst formation [2,3]. Apical periodontitis involves activation of both the innate and adaptive immune systems characterized by the recruitment of various types of cells and inflammatory mediators, which eventually leads to the destruction of periapical tissue and the formation of periapical lesions [4] (Figure 1). Therefore, apical periodontitis is the consequence of a complex interplay between microbiota of root canal system, microbial virulent factors and the host immune response [4].

## 2. Apical Periodontitis-A Global Burden

Apical periodontitis poses a significant global burden. The NHS in England and Wales have reported that over 1 million teeth received RCT between 2001 and 2004, costing the NHS around GBP 50.5 million [5]. According to the American Association of Endodontists, more than 15 million root canal treatments are performed each year [6]. In Europe, it is reported that almost 23 million endodontic treatments are undertaken yearly [7]. Furthermore, a systematic review reported a global high prevalence of apical periodontitis (5% of all teeth, one periapical lesion per patient) and root canal treatment (10% of all teeth, two root canal treatments per patient) in an adult population [8]. These numbers highlight that root canal treatment is considered one of the most common oral diseases, which could increase the burden on systemic health as well as increase the burden of costs on health services globally.

## 3. Focal Infection

Focal infection is defined as a localized or generalized infection caused by the systemic spread of bacteria or their products from distant foci of infection [9]. A focus of infection is “a confined area that is chronically infected with pathogenic microorganisms” [10]; it may be clinically asymptomatic and can occur anywhere in the body. In medical and dental literature, the teeth and oral tissues, tonsils, adenoids, etc., have all been cited as putative foci of infection [11,12]. The height of focal infection theory’s popularity was during the late 19th and early 20th century and was known as the ‘Focal Infection Era in Dentistry’. During this focal infection era, rheumatoid arthritis was closely associated with dental health. This resulted in widespread implementation of removal of teeth, adenoids, tonsils, and other organs for many decades, in an attempt to cure many unexplained illnesses which were allegedly caused by focal infection.

In 1891, Miller proposed that oral microorganisms and/or their by-products can spread to distinct body parts, drawing attention towards the relationship between oral and systemic disease [13]. Although Miller’s claims were not based on scientific grounds and were mostly from unsubstantiated anecdotal evidence and case reports, William Hunter proposed that oral microorganisms and their toxic by-products can spread from a focus of infection and cause a range of systemic conditions [14,15]. In 1925, Western Price advocated tooth extraction as the treatment of choice, believing that toxins and bacterial components produced by residual bacteria entrapped in dentinal tubules act as antigens (substances that are foreign to the host), and these antigens may travel through the bloodstream and lymphatic system to remote body parts and play an etiological role in causing many types of degenerative systemic diseases. However, Easlick (1952) pointed out flaws in Price’s methodologies and refuted any associations between endodontically treated teeth and systemic disease [9]. This subject laid dormant for decades due to lack of direct cause and effect evidence until Newman [16], again, brought this subject into attention. Since then, various studies have attempted to investigate whether endodontic disease, as a localized oral infection, could have an impact on the host immune response compromising the general health of individuals.

## 4. Endodontic Disease and Systemic Impact

Recently, there was a shift again in endodontics, from a discipline of pain management, infection control and tooth preservation toward oral infections as risks factors for systemic complications. The impact of apical periodontitis extends beyond its dental implications, e.g., tooth extraction (60–80% cases) [5]. There has been resurgence of the “Focal infection Theory” and this correlation between focal infection in the oral cavity and systemic diseases has again provoked global attention [17]. It can affect a patient’s health in terms of both the pathogenic effects of polymicrobial communities and the host immune responses [2]. Endodontic disease can result in translocation of microbes from the root canal into the systemic environment, triggering immune responses that can affect other tissues/organs. Studies have linked apical periodontitis with systemic diseases including diabetes [18], hypertension [19,20], adverse pregnancy outcomes [21], skeletal infections, and coronary heart disease (CHD) [22,23,24,25,26,27,28,29,30,31,32,33,34,35], the most common type of cardiovascular disease (CVD). This is due to increased risk of bacteraemia [36,37,38,39], translocation of soluble microbial compounds, active inflammatory mediators and haemostatic factors from the root canal into the systemic environment [40,41,42], resulting in metastatic infection, injury and inflammation, triggering low-grade systemic inflammation affecting other tissues and organs (Figure 2). Clinically, apical periodontitis can present completely asymptomatic and be detected as an accidental finding on an intraoral radiograph as a periapical radiolucency with no obvious signs and symptoms such as pain, swelling, abscess and sinus tract. So, not only symptomatic cases but also asymptomatic cases that remain unnoticed for years may have an adverse effect on a patient’s general health. Therefore, endodontic disease poses a major global health burden.

## 5. Similarities between Periodontal and Endodontic Disease Impacting Systemic Health

There is strong evidence in the literature correlating periodontal infections with increased risk of cardiovascular disease development [43,44,45,46,47]. A cross-sectional analysis of a large-scale study with a cohort of 60, 174 individuals after screening all patients’ records of 15 years concluded that there is an independent association of periodontitis with atherosclerotic cardiovascular diseases [48]. In a nationwide retrospective study, Byon et al. (2020) found that periodontitis can increase the risk of atherosclerotic cardiovascular disease, and its prevention may help in reducing the risk of cardiovascular disease [44]. Furthermore, the Consensus Report based on four papers [49,50,51,52] in a joint workshop organised by European Federation of Periodontology and American Academy of Periodontology in 2013 [53] concluded that there was strong and consistent epidemiological evidence that periodontitis results in increased risk of future atherosclerotic cardiovascular disease. This impact is biologically caused due to translocated circulating oral microorganisms, directly or indirectly inducing systemic inflammation resulting in the development of atherothrombogenesis [53]. Recently, a jointly organized workshop by the European Federation of Periodontology and the World Heart Federation in 2020 [54] concluded a latest consensus report that there was strong evidence that periodontitis patients exhibit significant prevalence of subclinical cardiovascular disease, heart failures and higher cardiovascular mortality (due to coronary heart disease and cerebrovascular disease). Other than cardiovascular diseases, studies have also linked periodontitis with type 2 diabetes mellitus, Parkinson disease, chronic obstructive pulmonary diseases, pneumonia, adverse pregnancy outcomes, osteoporosis, kidney disease, and most recently, the severity of COVID-19 [55,56,57,58,59,60,61,62,63,64]. It has also been found that periodontal disease and oral frailty in the elderly, and its interplay with oral microbiota, have a role in the diagnosis of different neurodegenerative diseases including Alzheimer’s disease [65,66,67]

While periodontal and endodontic disease have differences in their pathogenicity, they are both chronic infections and share common pathogens, inflammatory mediators [18] and biological pathways, thus linking these with systemic health [68]. Along with gingivitis and periodontitis, root canal infections also pose an increased risk of bacteraemia [36,37,38,39]. The anatomic proximity of these infections with the bloodstream can result in bacteraemia during treatment [39]. Moreover, in contrast to periodontal infections, no epithelial barrier is found between the necrotic infected root canal and highly vascular granulomatous tissue in periapical infections. In these lesions, areas of considerable bone resorption act as a “reservoir” of inflammatory biomarkers, including TNF alpha, IL6, IL-1β, PGE-2, and IL-8 [69,70]. Thus, endodontic disease is an enfolded primary infective focus for dissemination via periapical vasculature into the systemic circulation of either microbes, which can invade endothelial cells and promote a vascular inflammatory state, or the microbial by-products and localised inflammatory mediators that might trigger the immune response affecting other tissues and organs [36,37,38,39,40,41,42,71].

Attempts have been made to evaluate the effect of apical periodontitis on the development of systemic conditions including cardiovascular diseases (CVDs) [33,72,73], diabetes [34,35] and adverse pregnancy outcomes [74].

## 6. Endodontic Bacteraemia

Earlier studies have shown bacteraemia after root canal treatment [75,76]. These studies concluded that the possibility of bacteraemia increases when root canal instrumentation is performed beyond the root apex compared to when confined within the root canal system. Bender et al. (1960) showed that bacteraemia following root canal treatment is transient and lasts for up to 10 min after instrumentation, as the circulating microbes are cleared by the patient’s immune system [75]. Baumgartner et al. (1976) also showed that bacteraemia did not occur if root canal instrumentation was confined within the root canal [77].

Using a culture-based approach, studies demonstrated bacteraemia in around 3–20% cases after non-surgical root canal treatment [77,78]. However, most of the earlier studies had limitations with regard to the sensitivity of the blood culture techniques that they used. Debelian et al. (1992, 1995) published research work highlighting that bacteraemia is not only associated with overinstrumentation of the root canal beyond the apex but also even when instrumentation was maintained within the root canal system [39,79]. Furthermore, Debelian et al. (1995) using biochemical tests, and antibiograms established that the microorganisms isolated from the blood had the root canal as their source [39]. In subsequent studies using electrophoresis, DNA hybridization, and phenotypic and genetic methods, they further confirmed the endodontic origin of bacteraemia microorganisms [37,80,81]. Due to the higher sensitivity of the identification techniques employed in these studies, far greater bacteraemia (31% to 54%) was detected after root canal treatment than reported in the past.

Savarrio et al. (2005) also confirmed these results and identified bacteraemia by conventional culturing approach in 30% of the cases [38]. They also showed using pulsed field gel electrophoresis that microbes identified from blood and the root canal were genetically similar. Since more than half of the bacteria are unculturable, the relatively lower detection rate in the earlier studies after root canal treatment can be attributed to the use of a culture-based approach. Reis et al. (2016), using a molecular approach (qPCR), detected bacteraemia after non-surgical root canal therapy in all cases that were detected negative for bacteraemia with a culture approach [71] (Table 1).

Therefore, the incidence of bacteraemia is much higher than those reported in previous studies using a culture technique. The dissemination of microorganisms into the blood stream is common and can occur less than 1 min after an oral procedure. Microorganisms from the infected site may reach the lungs, heart, and peripheral blood capillary system [42,82] and contribute to the development of CVDs. Another well-known life-threating condition that can occur due to bacteraemia, especially in high-risk patients, is infective endocarditis. It is an infection of the heart lining, a heart valve or a blood vessel affecting 3.6 in 1,000,000 individuals per year. The patient can suffer from fever, heart murmurs, myocardial abscess, valvular incompetence, or mycotic aneurysm along with impacts on the central nervous system including stroke, transient ischemic attack, subarachnoid haemorrhage, brain abscess and toxic encephalopathy [83,84,85]. Therefore, bacteraemia associated with endodontic infections and treatment can have an adverse impact on general health.

## 7. Systemic Inflammatory Mediators

There is emerging evidence supporting that apical periodontitis can modify the systemic levels of inflammatory markers (e.g., high-sensitivity C-reactive protein (hs-CRP), Interleukin-1β (IL-1β), IL-6, IL-12, IL-10, tumour necrosis factor (TNF-α), matrix metalloproteinases (MMP-8 andMMP-9), soluble vascular cell adhesion molecule 1 (sVCAM-1), endothelial leukocyte adhesion molecule (E-selectin), and intercellular adhesion molecule (ICAM)), Immunoglobulin (Ig) A, IgM, IgG, asymmetric dimethylarginine (ADMA) and complement-C3 levels) in humans [86,87,88,89,90,91,92,93,94,95,96,97,98,99,100,101]. This can contribute to increased systemic inflammation. This is significant, not only for symptomatic apical periodontitis and failed root canal treated teeth, but also for highlighting the potential adverse impact of asymptomatic apical periodontitis on the systemic health and significance of successful endodontic treatment in protecting against that.

Interventional studies have shown significant differences in levels of inflammatory markers including CRP, C3 and ADMA between baseline and follow up [90,101,102,103,104]. In a longitudinal interventional study, Bakhsh et al. (2022) found that the pre-operative serum levels of IL-1β, hs-CRP, FGF-23, and ADMA were significantly higher in patients with apical periodontitis than healthy controls. This indicated the increased systemic burden associated with apical periodontitis. Furthermore, one year post treatment, the levels of these markers were generally reduced, indicating the positive effect of surgical and non-surgical root canal retreatment on the levels of these markers [105]. The reduction in these biomarkers after treatment seems to confirm the effectiveness of the available therapeutic approaches to endodontic treatment in suppressing systemic inflammation. This highlights the pathway of future research towards investigating the diagnostic potential of these biomarkers that can be used along with the current objective criteria (clinical and radiological) to assess endodontic success and also as a prognostic marker of systemic response to endodontic treatment.

## 8. Apical Periodontitis and Cardiovascular Diseases

“Cardiovascular diseases” (CVDs) is an umbrella term for conditions affecting the heart and blood vessels including coronary heart disease, cerebrovascular disease, stroke, hypertensive heart disease, cardiomyopathies and myositis, rheumatic heart disease, atrial fibrillation and flutter, congenital heart disease, valvular heart disease, peripheral artery disease, deep vein thrombosis, thromboembolic disease, and transient ischemic attack [106,107]. CVDs are a global health and economic burden as these are the leading cause of death worldwide, responsible for about 30% of total global mortality [108]. Furthermore, it is expected that the incidence of CVDs will increase by approximately 10% over the next 20 years, resulting in a threefold increase in healthcare cost [109].

Studies have suggested a relationship between endodontic infection and coronary heart diseases, the most common type of cardiovascular disease—49% of total CVD burden [22,23,24,106]. For example, in a hospital records-based study, An G.K. et al. (2016) [110] found that patients with apical periodontitis are 5.3-fold more likely to suffer from CVDs than those without apical periodontitis. The association was also evident in the study carried out by Virtanen et al. (2017) [111]. However, both studies included smoker patients, which is also a risk factor for CVDs.

In the past decade, the association between apical periodontitis and CVDs has been widely investigated. Since elevated inflammatory biomarker levels can induce a systemic inflammatory response, it may increase the risk of cardiovascular events [112,113,114]. The inflammatory response may also be associated with endothelial dysfunction [100,115], endothelial cell activation, and atherosclerosis [116]. Furthermore, studies have also investigated the impact of CVD on endodontic treatment outcome. A systematic review and meta-analysis of longitudinal cohort studies reported that patients with CVD have 67% risk of a negative endodontic outcome [117]. Since both CVD and endodontic disease can lead to inflammatory bioburden, disrupting the homeostasis and causing further impairment of the immune response may negatively impact endodontic treatment outcome.

Studies have shown a similarity between specific apical periodontitis inflammatory markers and the ones involved in atherosclerosis. In a systematic review and meta-analysis analysing the effect of apical periodontitis on levels of inflammatory mediators, Georgiou et al. (2019) found that apical periodontitis can increase the levels of CRP, IL-6, ADMA, and complement-C3 levels; however, the authors suggested the need for further well-controlled longitudinal studies [78]. Several systematic reviews and meta-analyses have been carried out to demonstrate associations between elevated levels of biomarkers in patients with apical periodontitis and the development and progression of CVDs [27,29,31,72,118]. Recently, Jakovljevic et al. (2020) [73], in an umbrella review, revealed that based on moderate to critically low-quality available evidence, the association between apical periodontitis and CVDs is weak, and the authors highlighted the need for future, well-designed, longitudinal clinical studies to strengthen the evidence to confirm a potential association.

## 9. Atherosclerosis

The major mechanism in the pathogenesis of coronary heart disease and cerebrovascular disease, which are the most frequent CVDs, is the development of atherosclerosis [119,120]. This is an inflammatory process that involves the formation of atherosclerotic plaque, affecting the tunica intima, tunica media, and tunica adventitia layers of large- and medium-calibre arteries, including the coronary artery [121,122]. These plaques are the accumulation of lipids and connective tissue along with inflammatory, endothelial, and smooth muscle cells [120]. Inflammation is regarded as the principal factor for the atherosclerotic plaque’s initiation, progression, and rupture, leading to thrombosis and its systemic complications, including myocardial infarction and stroke [123,124].

Endothelial dysfunction is caused by low-grade chronic inflammation triggered by pathogenic factors such as microorganisms or CVD risk factors, including high levels of low-density lipoproteins (LDL), hypertension, hyperlipidaemia, smoking-induced toxins, free radicals, shear stress, and/or a combination of these factors [125]. Endothelial dysfunction results in increased endothelial permeability, which allows migration of cholesterol-filled LDL into the vessel wall. The LDL particles then become oxidised and stimulate the release of phospholipids. As a result, an inflammatory response is elicited, and monocytes are attracted to the lesion, which becomes macrophages. These macrophages then engulf the oxidised LDL and transform into foam cells, which are precipitated into the vessel wall, resulting in the formation of fatty streaks [124,126]. There is upregulation of the vascular soluble adhesion molecules including ICAM-1, sVCAM-1 and E-selectin. These facilitate transmigration of monocytes and T-lymphocytes into the intima layer, resulting in the secretion of the pro-inflammatory cytokines including IL-1β, IL-6 and TNF-α. IL-6 triggers the release of C-reactive proteins (CRP) from hepatocytes. The release of these cytokines and growth factors eventually results in the formation of atheroma, which is a necrotic core composed of macrophages, lipid-laden cells, mast cells, T-cells and degenerative material covered by a thin fibrous cap [121]. As the process persists, the fibrous capsule is thinned, leading to plaque destabilisation and thrombus formation, which can block coronary, cerebral or peripheral blood vessels, resulting in myocardial infarction, stroke or peripheral arterial disease [120,121] (Figure 3).

There are several potential pathways by which chronic apical periodontitis could affect the development and progression of atherosclerosis. Firstly, endodontic microorganisms could directly seed in the arterial wall through bacteraemia, triggering a local inflammatory response including adaptive immune responses, inducing cellular alterations, which eventually results in the development of atherosclerotic plaques [127,128]. Secondly, damping of the endodontic bacterial by-products or local inflammatory mediators in the systemic circulation can lead to endothelial dysfunction and progression of the atherosclerotic inflammatory process [33]. Several studies have shown both bacteria and biomarkers of oral origin in atherothrombotic plaques or vascular biopsies [86,129,130]. Therefore, the presence of bacteraemia and the low-grade systemic inflammation associated with chronic apical periodontitis may contribute to the development of CVDs [87,131].

## 10. Inflammatory Mediators of Apical Periodontitis and Their Role in the Development of CVDs

### 10.1. C-Reactive Protein (CRP)

CRP belongs to the pentraxin family [132]. The hepatocytes synthesise it in response to IL-6 [133]. CRP is considered a non-specific systemic inflammatory biomarker and is widely used to monitor infections and inflammatory conditions [134]. CRP, by enhancing inflammation, oxidative stress, and coagulation, is involved in various steps leading to vascular events [135].

CRP can activate complement C3, upregulate vascular adhesion molecules, trigger proinflammatory cytokines (IL-1 and TNF-α), recruit monocytes into the arterial wall, and cause superoxide, myeloperoxidase, and matrix metalloproteinases elevation. It can damage endothelial vasoreactivity and facilitate low-density lipoprotein uptake by endothelial macrophages to form foam cells [124,135,136].

Several investigations have associated elevated levels of CRP with future cardiovascular events, including acute myocardial infarction (MI), stroke, and peripheral artery disease [137]. Indeed, a hs-CRP has been suggested as a screening biomarker to evaluate coronary heart disease risk [138,139]. It is also strongly correlated with several cardiovascular risk factors, including diabetes, obesity, hypertension, and lipids [140,141]. However, CRP has its limitations as it is a non-specific marker and its levels can dramatically increase in cases of infection and tissue damage [142].

In a recent study, it was found that poor oral health, periodontal disease and tooth loss were associated with higher levels of CRP, which may be an indicator of the contribution of periodontal disease to chronic systemic inflammation, and can also be a contributor towards the progression of atherosclerosis and thrombus formation [143]. A previous histological study reported increased IL-6 and CRP messenger RNA levels in periodontal ligament tissue of teeth with apical periodontitis [144]. Enhanced CRP synthesis, in response to IL-6 in apical periodontitis, can act as a potential reservoir of IL-6 and CRP for sustaining a low systemic inflammatory response [144,145,146], thus increasing the risk for atherosclerotic cardiovascular disease. Furthermore, Vidal et al. (2016) showed that apical periodontitis was associated with higher CRP levels in plasma of hypertensive patients [147]. Garrido et al. (2019) also reported higher serum levels of hs-CRP in individuals with apical periodontitis when compared to healthy controls [96]. Sirin et al. (2019) also showed a positive correlation of increased serum hs-CRP levels with increasing severity of apical periodontitis [89]. On the other hand, the impact of root canal treatment on systemic levels of hs-CRP was tested in a study conducted by Poornima et al. (2020). The study results demonstrated that root canal treatment has a positive impact in reducing the levels of hs-CRP in systemically healthy patients with apical periodontitis [104]. However, when investigating a larger sample size, Bakhsh et al. found that surgical and non-surgical root canal retreatment initially increased the serum levels of hs-CRP within 3 to 6 months after treatment but the levels declined at the one year review [105].

### 10.2. Pentraxin-3 (PTX-3)

PTX-3 is a member of the long pentraxin family. It is expressed at sites of inflammation by several cells including stromal (endothelial cells, fibroblasts), myeloid cells (monocytes/macrophages), polymorphonuclear neutrophils in a response to the primary proinflammatory stimuli (IL-1β, TNF-α), bacterial LPS, flagellin, outer membrane protein, and ischaemia [148,149]. Studies have shown that increased levels of PTX3 increase the risk of cardiovascular diseases [149,150,151]. Pentraxin 3 is also involved in atherosclerosis by interacting with many ligands and acts as a modulatory molecule of the complement system, inflammatory response, angiogenesis, and tissue remodelling [152]. It has been found that levels of PTX3 are useful in indicating local inflammation at atherosclerotic lesions more accurately than CRP. This marker was investigated for the first time in apical periodontitis patients by Bakhsh et al. (2022). The study showed that serum levels of PTX-3 significantly reduced at one year after surgical and non-surgical root canal retreatment [105]. This indicates that system inflammatory burden of PTX-3 can be raised in patients with apical periodontitis, whereas endodontic treatment has a positive effect of on PTX-3 serum inflammatory levels.

### 10.3. Asymmetric Dimethylarginine (ADMA)

ADMA is an analogue of L-arginine that occurs naturally in plasma. It is an endogenous inhibitor of nitric oxide (NO) synthase, which catalyses the production of nitric oxide. NO modulates vascular tone, endothelial function and has a biological effect, especially in the cardiovascular system [153,154,155]. Therefore, the increased ADMA levels by inhibiting nitric oxide synthase and NO would result in endothelial dysfunction associated with atherosclerosis [156,157]. Inflammatory stimuli can result in increased ADMA levels, which subsequently increases the risk of coronary heart disease [158,159].

In a clinical study conducted to assess whether patients with apical periodontitis were at risk of developing an atherosclerotic lesion, Cotti et al. (2011) found that patients with apical periodontitis had significantly higher levels of ADMA, and significant reduction in endothelial flow reserve when compared to controls [100]. Additionally, Georgiou et al. (2019) found in their systematic review that apical periodontitis increases the systemic levels of ADMA when compared to controls [87].

Bakhsh et al. (2022) also found that the pre-operative serum levels of ADMA were significantly higher in patients with apical periodontitis compared to the controls. ADMA serum levels were reduced at one year post endodontic treatment; however, the increased ADMA levels at baseline caused a significant reduction in the proportion of successful outcomes [105].

### 10.4. Fibroblast Growth Factor-23 (FGF-23)

FGF-23 is a hormone produced by osteocytes and osteoblasts that increases the activity of the kidneys to metabolise phosphate and vitamin D. Any inflammatory bone alteration indirectly impacts the production of FGF-23 [160,161]. Furthermore, several studies have found that FGF-23 is also regulated by LPS, IL-1β and TNF-α [162]. Higher levels of FGF-23 were found to have an impact on the kidney and the heart. In the kidney, high levels of FGF-23 would cause an increase in sodium absorption and renin–angiotensin activation, which would subsequently lead to hypertension. Moreover, heart and blood vessels are affected by high levels of FGF-23, which could lead to subclinical atherosclerosis, cardiovascular events, left ventricular hypertrophy, and death [163,164]. Bakhsh et al. (2022) investigated serum FGF-23 levels in apical periodontitis patients and found significantly higher levels at the baseline compared to control. FGF-23 levels at the baseline were also positively correlated to the preoperative size of the periapical radiolucency. Interestingly, the levels of this marker reduced at every subsequent review appointment with significant reduction at 1 year post surgical and non-surgical root canal retreatment [105]. This highlights the FGF-23 system inflammatory burden caused by apical periodontitis and the positive effect of endodontic treatment on its reduction.

### 10.5. Matrix Metalloproteinases (MMPs)

Matrix metalloproteinases are enzymes that are involved in the physiological and pathophysiological processes of tissue repair and remodelling. They are stimulated by pro-inflammatory cytokines (IL-1β and TNF-α) and maintain a persistent inflammatory process in the periapical region when released. MMP-1, MMP-2, MMP-3, MMP-8, MMP-9, and MMP-13 have been shown to be present in periapical lesions from humans [165,166,167,168,169,170,171,172,173]. Furthermore, MMPs play a significant role in several pathological diseases including atherosclerosis and early development of hypertension [174,175]. Increased proteolytic activity of MMPs results in atherosclerotic plaque ruptures leading to cardiovascular events [176,177,178,179]. MMP-2 secreted by fibroblasts in primary endodontic infections aids in the periapical inflammation and tissue destruction [180]. MMP-8 is a neutrophil collagenase and, during inflammation, degrades collagen types I, II, and III. It is activated by autolytic cleavage, and its upregulation was found in inflamed pulp and periradicular lesions. Pattamapun et al. (2017) found both MMP-2 and MMP-8 in root canal exudates and their levels gradually decreased upon root canal treatment, suggesting that MMPs play a role in the healing of periapical lesions [181].

### 10.6. Human Complement C3

Complement C3 is a protein complex of the innate immune system. Both intrinsic and extrinsic stimuli play a role in the activation of C3. This results in recruitment of phagocytes and target cell lysis [182]. Acute-phase reactant C3 fragment is linked with several systemic conditions, including metabolic syndrome, diabetes mellitus, smoking, and atherosclerotic CVD [182,183]. In addition, it has been observed that increased serum levels of C3 are associated with increased risk of CVDs [182,183]. Studies have demonstrated a reduction in the levels of C3 after endodontic treatment, thus confirming the effectiveness of endodontic treatment in suppressing systemic inflammation.

Kettering and Torabinejad (1984), looking at the effect of dental abscess on the levels of C3, found that serum levels of C3 were higher in patients with acute apical conditions when compared to controls. These levels were reduced following root canal treatment and extraction [101]. Furthermore, Márion et al. (1988) investigated C3 levels in patients with chronic periapical granuloma and found similar results following periapical surgery [102]. More recently, in a systematic review and meta-analysis, Georgiou et al. (2019) found that the presence of apical periodontitis contributed to the elevated levels of C3. Root canal treatment resulted in reduced levels of C3, which could help reduce the risk of CVDs [87].

### 10.7. Statins and Apical Periodontitis

Elevated triglycerides and LDL and low high-density lipoprotein (HDL) levels are known risk factors for the development of atherosclerosis and CVDs. Several studies found a positive association between periodontitis and increased triglyceride levels [184,185]. Statins are a group of medicines that can help lower the levels of LDL cholesterol in the blood and are administered in patients with hypercholesterolaemia with associated increased risk of atherosclerosis and heart diseases, including coronary heart disease and risk of cardiac infarction [186]. This medication has pleiotropic effects such as increased osteoblastic differentiation [187,188,189], promotion of viability and proliferation of osteoblasts [190,191] and improvement of mineralization [192,193,194,195]. Statins also inhibit osteoclastogenesis through their effect on the RANKL-induced nuclear factor kappa β (NF-κβ) activation pathway [196]. In periodontitis patients taking statins, a conjoint benefit was revealed with scaling and root planning [197].

Statins’ effect on apical periodontitis healing has also been investigated. In an animal study, Lin et al. (2009) found that the introduction of simvastatin before the induction of periapical lesion significantly reduced bone resorption when compared to the control group [198]. This is due to the anti-inflammatory and immunomodulatory effect of statin by decreasing CD-68-positive macrophages and the protection of osteoblast [198]. In another animal study, Pereira et al. (2016) also showed that the use of simvastatin decreased the progression of increasing periapical ligament space in apical periodontitis-induced rats [199]. Alghofaily et al. (2018) tested the effect of long-term statin intake on the healing of apical periodontitis and found that there was a significant association between long-term statin intake and healing of apical periodontitis after non-surgical root canal treatment [196]. Although these studies provide some evidence of the positive effect of statins on the healing of apical periodontitis, further investigations are required to establish this fact.

## 11. Apical Periodontitis and Diabetes Mellitus

Diabetes mellitus (DM) is a complex multisystem metabolic syndrome characterised by abnormalities in carbohydrate, protein and lipid metabolism due to either profound or an absolute insulin deficiency caused by pancreatic β-cell dysfunction (type 1) and/or insulin resistance in liver and muscle (type 2) [200]. DM can affect the immune system of the individual by upregulation of pro-inflammatory cytokines from monocytes and polymorphonuclear neutrophils along with downregulation of growth factors from macrophages, which predisposes them to chronic inflammation, progressive degradation of tissues and diminished tissue repair capacity (Figure 4) [201]. Diabetes can eventually lead to dysfunction of several organs such as the kidneys, nerves, eyes, blood vessels and the heart. It has been reported that diabetes is associated with increased morbidity and mortality [202,203]. DM is a global health burden; in 2019, DM was affecting around 463 million adults. It is expected that these figures could reach around 700 million by year 2045 [204].

Chronic systemic inflammation in DM causes an alteration and elevation in the serum levels of proinflammatory markers TNF-α, IL-1α, IL-1β, CRP and IL-6 [205,206], which can have a negative impact on periapical healing [207]. Systemically, DM inhibits collagen formation and alters the degeneration of matrix proteins and tissue remodelling, which leads to poor wound healing [208]. Garber et al. (2009) showed poor wound healing with direct pulp capping using mineral trioxide aggregate (MTA) in diabetic rats. The results also showed lower dentin bridge formation and elevated pulpal inflammation [209]. There is strong evidence from the literature that patients with DM have higher prevalence of apical periodontitis, greater periapical lesion size and greater incidence of periapical infections as compared with patients who do not have diabetes [210,211,212,213,214]. In a retrospective study, Segura-Egea et al. (2005) showed a higher prevalence of untreated periapical lesions and unsuccessful endodontic treatment in patients with DM [212]. There was a trend toward increased symptomatic periradicular disease in patients with diabetes who received insulin, as well as flare ups in all patients with diabetes [210,211,212,213].

On the other hand, the results of some studies suggest that chronic periapical disease correlates with higher HbA1C levels and contributes to diabetic metabolic dyscontrol [26,215]. The inflammatory periapical response is enhanced in diabetics, leading to a rise in blood glucose with intensification of diabetes, requiring an increase in insulin dosage or therapeutic adjustment [216]. Yip et al. (2021) provided evidence linking DM and the level of glycemia to the increased prevalence of apical periodontitis. The study also implied that statins and metformin use may be protective in this relationship as they were associated with lower prevalence of apical periodontitis [214].

Furthermore, the available scientific evidence strongly suggests that DM has a negative impact on the outcome of endodontic treatment in terms of periapical healing due to delay or arrest of periapical repair. Ng et al. (2011) found that DM is one of the prognostic factors for the survival of root-filled teeth [217]. There is a decrease in the success of endodontic treatment in cases with preoperative periradicular lesions in patients with DM. The prognosis for root-filled teeth is worse in diabetics, showing a higher rate of root canal treatment failure with increased prevalence of persistent chronic apical periodontitis [210,211,212,213]. Therefore, diabetes contributes to decreased retention of root-filled teeth and is a significant risk factor for tooth extraction after non-surgical root canal treatment [20,218].

Since diabetes is the third most prevalent chronic medical condition in patients seeking dental treatment [216], dentists should be aware of the possible relationship between diabetes and endodontic infections. Diabetic patients, especially those with poor glycaemic control, should be informed about the evidence of poor outcome of endodontic treatment with increased risk of failure associated with diabetes. This should be part of informed consent and also care planning should include liaising with the patient’s physician.

## 12. Apical Periodontitis and Pregnancy

Periodontal diseases have been shown to burden pregnant patients due to systemic inflammatory stress [219]. Studies indicated that Prostaglandin E2 (PGE-2) and TNF-α from inflamed periodontal tissues in pregnant women can reach the placenta and amniotic fluid, contributing to preterm birth [219,220,221,222]. Recently, the association between apical periodontitis and adverse pregnancy outcomes has also been investigated. Studies showed that the presence of a periapical lesion in postpartum women was associated as a risk factor for shorter pregnancy duration, intrauterine growth restriction and preterm birth [223,224]. Khalighinejad et al. (2017) found that maternal apical periodontitis may be a strong independent predictor of preeclampsia [225]—the most common adverse pregnancy outcome characterized by hypertension and proteinuria after the 20th week of gestation [226]—and is among the leading causes of maternal mortality [227]. In a recent systemic review, Jakovljevic et al. 2021 critically evaluated the available evidence on the association of maternal apical periodontitis with adverse pregnancy outcomes. The authors concluded that based on ‘Fair’ and ‘Good’ quality available evidence, a positive association was observed between maternal apical periodontitis and adverse pregnancy outcomes [74]. Therefore, it could be suggested that the risk of preeclampsia and low birth-weight preterm birth may be reduced through timely diagnosis and treatment of any source of inflammation, including apical periodontitis, before pregnancy.

## 13. Apical Periodontitis and Autoimmune Disorder

Autoimmune disorders are a group of conditions that share a self-reactive immune response involving different inflammatory mediators [228]. Inflammatory Bowel Diseases (IBD) including Ulcerative Colitis and Crohn’s Disease [229], along with Rheumatoid Arthritis (RA) and Psoriasis (Ps) are examples of autoimmune disorders. Studies have shown a higher prevalence of apical periodontitis in some autoimmune disorders such as IBD and RA [230,231,232]. Recently, Ideo et al. (2022) showed similar findings where patients affected by autoimmune diseases (RA, Ps and IBD) had a higher prevalence of apical periodontitis compared to the controls [233]. This may be attributed to the role of excessive production of common inflammatory cytokines such as TNF-α, IL-1, IL-6, IL-23 and IL-17 in the development, progression and persistence of both conditions [2,234,235,236]. Furthermore, the RANKL osteoprotegerin (OPG) pathway is involved in the progression of RA as well as apical periodontitis [235].

Immune system status plays an essential role in the development and progression of apical periodontitis. The medications used for the treatment of these autoimmune disorder modify the immune response and include conventional Disease-Modifying Anti-Rheumatic Drugs (cDAMRDs) [237,238,239] and biologic Disease-Modifying Anti-Rheumatic Drugs (bDMARDs) [240,241,242]. bDMARDs block targets’ activity in the inflammatory process including cytokines (TNF-α, IL-6, and IL-1); RANKL-induced nuclear factor kappa β activation pathway; and T or B cell receptors [243,244]. Piras et al. (2017) found that the frequency of teeth with apical periodontitis was significantly higher in patients with autoimmune disorders receiving bDMARDs [230]. In a recent study, Ideo et al. (2022) showed similar results where patients with autoimmune diseases taking biologic medications had a higher prevalence of apical periodontitis [233]. Cotti et al. (2015, 2018) showed that endodontic treatment of teeth with apical periodontitis in patients taking biologic medications resulted in faster healing than among controls, thereby suggesting that immune-modifying treatment may influence the healing of apical periodontitis after treatment [245,246].

Therefore, patients with autoimmune disorder due to altered immune response and influence of immune modulatory therapy can have an impact on the prevalence of apical periodontitis and prognosis after endodontic treatment.

## 14. Apical Periodontitis and Other Systemic Conditions

Although it is not yet confirmed, some researchers have tried to correlate the presence of apical periodontitis with different systemic conditions including liver diseases and haemophilia [247,248]. In a cross-sectional study, Castellanos-Cosano et al. investigated the frequency of apical periodontitis among patients undergoing liver transplant assessment and found that 79% of the study participants had one or more apical periodontitis when compared to healthy controls [247]. Furthermore, the same group of authors investigated the prevalence of apical periodontitis in patients with inherited haemophilia and found that an apical radiolucency was present in almost 68% of patients with haemophilia [248]. The findings of these investigations highly suggest that apical periodontitis is found in several systemic diseases which mandate the frequent dental follow-up and reinforcement of oral hygiene regime in medically compromised patients, not only to improve and maintain their oral health but also to decrease the systemic burden of oral disease in these patients.

## 15. Conclusions

There is emerging evidence that bacteraemia and low-grade systemic inflammation associated with chronic apical periodontitis may contribute negatively to systemic health such as the development of CVDs, adverse pregnancy outcomes, and diabetic metabolic dyscontrol. Although the evidence is limited, it supports that patients who have conditions such as DM or autoimmune disorders have an impact not only on the prevalence of apical periodontitis but also on the prognosis after endodontic treatment. Statins used may be protective in this relationship by having a positive effect on apical periodontitis healing. Furthermore, the convincing evidence supports that successful root canal treatment has a beneficial impact on systemic health by reducing the inflammatory burden, thereby dismissing the misconceptions rooting back to research performed 70–80 years ago about the relationship between endodontic treatment and focal infection, which resulted in arguments in favour of tooth extraction. Further high-quality research is required to strengthen this available evidence showing the benefits of endodontic treatment on systemic health.

## Figures and Tables

**Figure 1 medicina-58-00931-f001:**
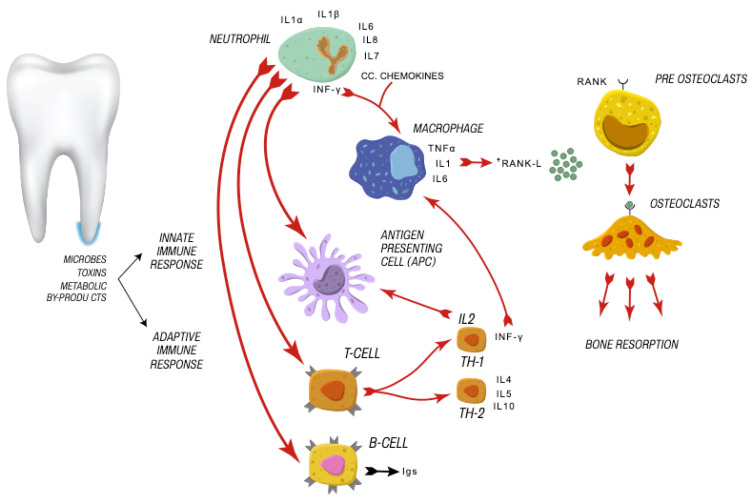
Cellular immune response activated by microbes, toxins, and metabolic by-products.

**Figure 2 medicina-58-00931-f002:**
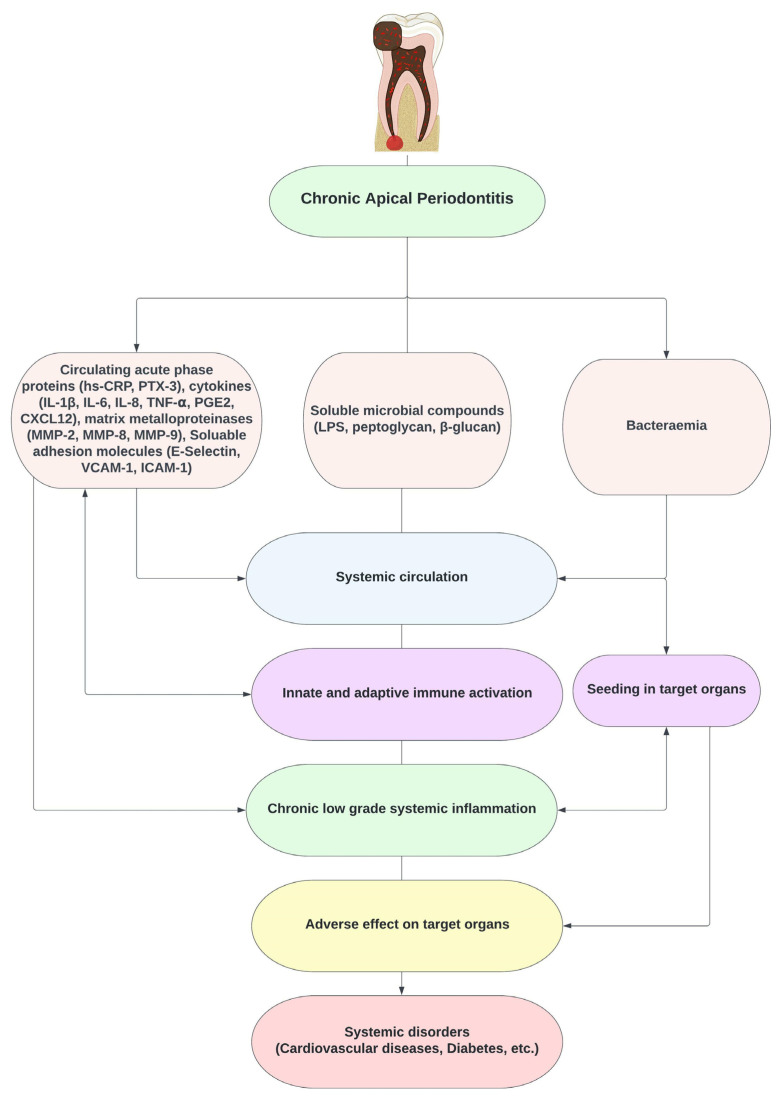
Pathways linking endodontic bacteria to systemic diseases.

**Figure 3 medicina-58-00931-f003:**
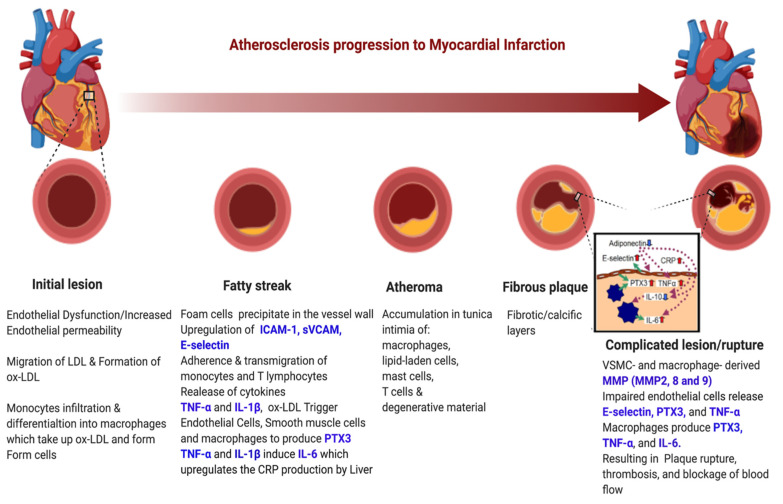
Atherosclerosis progression process. (Figure created using Biorender.com) (accessed on 16 June 2022).

**Figure 4 medicina-58-00931-f004:**
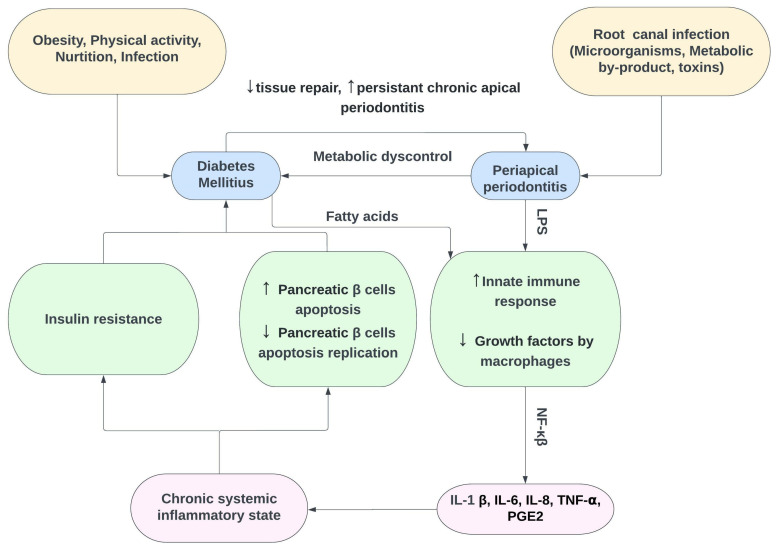
Interactions between diabetes mellitus and chronic periapical periodontitis. LPS: lipopolysaccharide, NF-κβ: nuclear factor kappaβ, PGE2: Postaglandin-E2, ↑ Increase, ↓ Decrease.

**Table 1 medicina-58-00931-t001:** Studies investigating bacteraemia associated with endodontic treatment.

Study	Method Used	Findings
[77]	Culture-based approach	3.3% after non-surgical root canal treatment 33% in periapical curettage case 83.3% in surgical flap reflection 100% after tooth extraction
[78]	Culture-based approach	100% after tooth extraction 70% after dental scaling 20% following root canal treatment
[79]	Culture-based approach	*Propionibacterium acnes* was recovered from the blood in cases where overinstrumentation occurred
[39]	Culture-based approach	*Propionibacterium acnes*, *Peptostreptococcus prevotii*, *Fusobacterium nucleatum*, *Prevotella intermedia* and *Saccharomyces cerevisiae* were recovered from the blood in cases where overinstrumentation occurred *P. intermedia*, *Actinomyces israelii*, *Streptococcus intermedius* and *Streptococcus sanguis* were found in cases where instrumentation ended inside the canal
[80]	Culture-based approach using sodium dodecyl sulfate-polyacrylamide gel electrophoresis	Findings confirmed that the isolates identified from blood in previous studies including (*Propionibacterium acnes*, *Peptostreptococcus prevotii*, *Fusobacterium nucleatum*, *Prevotella intermedia*, *Actinomyces israelii*, *Streptococcus intermedius*, and *Streptococcus sanguis*) originated from the root canal
[81]	DNA-Hybridization	Results suggested that bacteria isolated from the blood originated from the root canal
[37]	Phenotypic and genotypic approach	All root canals contained anaerobic bacteria Frequency of bacteraemia varied from 31% to 54%
[38]	Culture-based approach	Bacteraemia found in 30% of the cases
[71]	Molecular approach (qPCR)	Detected bacteraemia after non-surgical root canal therapy in all cases that were detected negative for bacteraemia with a culture approach

## Data Availability

Not applicable.

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
