# Peer review of "Association between Endodontic Infection, Its Treatment and Systemic Health: A Narrative Review"

_medicina, 2022, doi:10.3390/medicina58070931_

Round 1

Reviewer 1 Report

This is a well-written, methodologically sound review on an extremely important yet in clinical practice often neglected topic: the association between endodontic infection and systemic diseases.

While the manuscript is certainly worthy of publication, also to raise attention on this important topic, there are some issues the Authors need to address.

Please find below my considerations.

1. There is a typo in the title (it’s treatment). Please, correct it

2. Lines 24-27: “Although the evidence is limiting… evidence is limited”. 

This paragraph needs to be rephrased because it seems a little bland and redundant.

3. Line 24 “the evidence is limiting”; line 27 “evidence is limited”; line 27 “convincing evidence”; line 29 “compelling evidence”; line 31 “strength this evidence”. 

The abstract needs to be rephrased to improve the understandability of the text.

4. A better justification is needed for this review since others have already been published on this topic even recently. Please see: Khalighinejad N. et al., J Endod. 2016 Oct;42(10):1427-34. doi: 10.1016/j.joen.2016.07.007; Segura-Egea JJ et al., Int Endod J. 2015 Oct;48(10):933-51. doi: 10.1111/iej.12507.

Leaving aside the differences between a narrative review and a systematic review, when the subject matter is the same, it is necessary to point out the strengths of the study the Authors intend to bring to the attention of the scientific community compared with what has been published before.

5. Lines 37-39: “Endodontic infection is a polymicrobial infection… other parts of the body. 

Where is/are the reference(s)?

6. The paragraph “Apical periodontitis – a global burden” Is very interesting and deserves to be implemented.

7. On the contrary, the whole part about the historical excursus regarding focal infection is of no help or support to the purpose of the review, which is instead to focus on the association between endodontic infection and systemic diseases (or at most summarized in very few lines).

8. From the perspective of research outlook, the oral microbiota deserves further consideration. 

9. Why did the Authors investigate the association between endodontic infection and some systemic diseases while leaving out others? (Chronic liver disease, inherited coagulation disorders, etc.)

10. Lines 159-161: why do the Authors cite only one cross-sectional study (despite the more than satisfactory sample examined), many longitudinal studies have studied the association between periodontal disease and cardiovascular diseases, providing a much higher degree of evidence.

11. Lines 169-171: “studies have also linked periodontitis with type 2 diabetes… and most recently, the severity of COVID-19”. 

In recent years great importance has also been given to the relationship between periodontal disease and oral frailty in the elderly, and its interplay with oral microbiota in the determinism of various neurodegenerative diseases, including, in particular, Alzheimer's disease. Please see: Dibello V. et al., Neural Regen Res. 2021 Nov;16(11):2149-2153. doi: 10.4103/1673-5374.310672; Borsa L. et al., Int J Environ Res Public Health. 2021 Sep 3;18(17):9312. doi: 10.3390/ijerph18179312; Dibello V. et al., Lancet Healthy Longev. 2021 Aug;2(8):e507-e520. doi: 10.1016/S2666-7568(21)00143-4

12. Line 193: “…only up to only 10 min after instrumentation…” Please, rephrase.

13. Lines 197-209: once again, it would be appropriate to focus more on studies carried out in recent years rather than research conducted since the second half of the last century (or at least try to streamline the text a lot).

14. Finally, the challenges of routine oral health assessments should be critically discussed.

Reviewer 2 Report

·    Title: Association between Endodontic infection, it’s Treatment and Systemic Health: A Narrative Review

  Overall the manuscript is well written. However, a few corrections need to be addressed.

·      Certain aspects of the manuscript seem to be elaborate and need to be re-written concisely for the ease and understanding of the reader.

·      Incorporation of more flowcharts and diagrams is suggested rather than detailed explanations of the concepts.

·      The reference count for the manuscript has exceeded and it requires modification. (please refer to journal guidelines for the same)

·      The word count for the abstract has exceeded the journal’s guidelines. (Please refer to journal guidelines for the same)

Page 8, line number 287: Grammatical error in the sentence is highlighted and needs to be corrected
